# PHYSICS-INFORMED KOOPMAN NETWORK FOR TIME-SERIES PREDICTION OF DYNAMICAL SYSTEMS

## ABSTRACT

Koopman operator theory is receiving increased attention due to its promise to linearize nonlinear dynamics. Neural networks that are developed to represent Koopman operators have shown great success thanks to their ability to approximate arbitrarily complex functions. However, despite their great potential, they typically require large training data-sets either from measurements of a real system or from high-fidelity simulations. In this work, we propose a novel architecture inspired by physics-informed neural networks, which leverage automatic differentiation to impose the underlying physical laws via soft penalty constraints during model training. We demonstrate that it not only reduces the need of large training data-sets, but also maintains high effectiveness in approximating Koopman eigenfunctions.

## 1 INTRODUCTION

Nonlinear dynamical systems give rise to a rich diversity of complex phenomena such as the ones arise in climate science Lorenz (1956), neuroscience Brunton et al. (2016), ecology Clark & Luis (2020), finance Mann & Kutz (2016), and epidemiology Proctor & Eckhoff (2015). In 1931, Koopman introduced the operator-theoretic perspective of dynamical systems , complementing the traditional geometric perspectives Mezic (2020). In this framework, a Koopman operator is defined which acts on observation functions (observables) in an appropriate function space. Under the action of this operator, the evolution of the observables are linear although the function space may be infinite-dimensional. As a consequence, approximating the Koopman operator and seeking its eigenfunctions become a key to linearize the nonlinear dynamics Mezic (1994); Mezić (2005); Brunton et al. (2021).

While there are various conventional methods to approximate Koopman operator spectra, e.g. DMD method Rowley et al. (2009) and its variants, or data-driven autoencoder-based methods, e.g. Takeishi et al. (2017); Morton et al. (2018); Lusch et al. (2018); Gin et al. (2021), they all face various challenges. DMD-based methods, for instance, require an a priori, judicious selection of the observables with no guarantee that these observables span an invariant Koopman subspace Kutz et al. (2016). Conversely, neural-network-based methods need to acquire a large enough data-set, which pose a challenge, e.g. in terms of computational cost or experimental complexities. In this work, we propose physics-informed Koopman networks (PIKNs) which assimilates the knowledge of the dynamical system and autoencoder-based Koopman networks. We demonstrate doing so reduce the need for large training data-sets and it enables the model to better predict beyond the training horizon. It should be noted that our contribution is the addition of physics-informed loss, which can be combined with any state-of-the-art data-driven approximation of the Koopman operator.

## 2 METHOD

### 2.1 KOOPMAN OPERATOR THEORY

For an autonomous ordinary differential equation of $\frac{d}{dt}\mathbf{x}(t) = \mathbf{f}(\mathbf{x}(t))$, with $\mathbf{x} \in \mathcal{X} \subseteq \mathbb{R}^n$, we define the time-$t$ flow map operator $\mathbf{F}^t : \mathcal{X} \to \mathcal{X}$ as

$$\mathbf{x}(t_0 + t) = \mathbf{F}^t(\mathbf{x}(t_0)) \tag{1}$$

In Koopman framework, an alternative description for dynamical systems is in terms of evolution of functions of possible measurements $g : \mathcal{X} \to \mathbb{C}$, which belongs to a set of functions $\mathcal{G}(\mathcal{X})$ of significantly higher dimension than $\mathcal{X}$. The family of Koopman operators $\mathcal{K}^t : \mathcal{G}(\mathcal{X}) \to \mathcal{G}(\mathcal{X})$, parameterized by $t$ are given by

$$\mathcal{K}^t g(\mathbf{x}) = g(\mathbf{F}^t(\mathbf{x})) \tag{2}$$

It can be shown $\mathcal{K}^t$, which in general an infinite dimensional, is linear. Constructing finite-dimensional representations of Koopman operator remains an open question, and is in the scope of this work. If $\mathbf{f}$ is sufficiently smooth, one can also define the infinitesimal generator $\mathcal{L}$ of the Koopman operator family as

$$\mathcal{L}g := \lim_{t \to 0} \frac{\mathcal{K}^t g - g}{t} = \lim_{t \to 0} \frac{g \circ \mathbf{F}^t - g}{t} \tag{3}$$

From the definition, we can easily see

$$\mathcal{L}g(\mathbf{x}(t)) = \lim_{\tau \to 0} \frac{g(\mathbf{x}(t + \tau)) - g(\mathbf{x}(t))}{\tau} = \frac{d}{dt} g(\mathbf{x}(t)) \tag{4}$$

The generator $\mathcal{L}$ is sometimes referred to as the Lie operator. On the other hand, we also have

$$\frac{d}{dt} g(\mathbf{x}(t)) = \nabla g \cdot \frac{d}{dt} \mathbf{x}(t) = \nabla g \cdot \mathbf{f}(\mathbf{x}(t)) \tag{5}$$

Therefore, we conclude

$$\mathcal{L}g = \nabla g \cdot \mathbf{f} \tag{6}$$

Equation 6 will be the key for the implementation of PIKN.

It is straightforward to show that Koopman eigenfunctions $\varphi(\mathbf{x})$ that satisfies $\mathcal{K}^t \varphi(\mathbf{x}) = \lambda^t \varphi(\mathbf{x})$ for $\lambda^t \neq 0$ are also eigenfunctions of the Lie operator, which we denote as $\phi_i$. It can be shown that

$$g(\mathbf{x}) = \sum_{k=1}^{M} c_k \varphi_k(\mathbf{x}) \implies \mathcal{K}^t g(\mathbf{x}) = \sum_{k=1}^{M} c_k \lambda_k^t \varphi_k(\mathbf{x}) \tag{7}$$

This also implies $span\{\varphi_k\}_{k=1}^{M}$ is an invariant subspace under the Koopman operator $\mathcal{K}^t$ and can be viewed as the new coordinates on which the dynamics evolve linearly.

Since our ultimate goal is to study nonlinear dynamical systems using linear theory, we do not need to restrict ourselves to Equation 7. Following Lusch et al. (2018); Gin et al. (2021), we can generalize it as

$$g(\mathbf{x}) = \psi(\varphi_1(\mathbf{x}), \varphi_2(\mathbf{x}), \dots, \varphi_M(\mathbf{x}); \omega)$$
$$\Downarrow \tag{8}$$
$$\mathcal{K}^t g(\mathbf{x}) = \psi(\lambda_1^t \varphi_1(\mathbf{x}), \lambda_2^t \varphi_2(\mathbf{x}), \dots, \lambda_M^t \varphi_M(\mathbf{x}); \omega)$$

where $\psi$ is an arbitrary transformation parameterized by $\omega$.

## 2.2 Physics-informed Koopman network

In physical sciences, data is scarce while governing equations are available in literature. In physics-informed Koopman networks (PIKNs), we aim to leverage such knowledge of the dynamics, e.g., of Equation 6, to enforce the linearity constraint. The basic idea is to train the network by minimizing the quantity $\|\nabla \varphi_k(\mathbf{x}) \cdot \mathbf{f} - \mu_k \varphi_k(\mathbf{x})\|$. More generally, a squared matrix $L$ is used to approximate the Lie operator $\mathcal{L}$, which in turn is related to the Koopman operator, and we minimize $\|L\phi(\mathbf{x}) - \nabla \phi(\mathbf{x}) \cdot \mathbf{f}\|$. Finding the eigenvalue and eigenfunction pairs of the Lie operator corresponds to performing eigendecomposition to the matrix $L$.

We can seamlessly integrate information from measurement data $X_{data} := \{\mathbf{x}(t_0), \mathbf{x}(t_1), \cdots, \mathbf{x}(t_p)\}$, and the knowledge (collocation points) $\bar{x} := \{\bar{x}(t_0), \bar{x}(t_1), \cdots, \bar{x}(t_p)\}$ such that the loss function is

$$\mathcal{J} = \frac{1}{N} \sum_{i=1}^{N} (\omega_1 \|L\phi(\bar{\mathbf{x}}_i) - \nabla \phi(\bar{\mathbf{x}}_i) \cdot \mathbf{f}(\bar{\mathbf{x}}_i)\|^2 + \omega_2 \|\bar{\mathbf{x}}_i - \psi(\mathbf{z_i})\|^2) +$$
$$\frac{1}{p} \sum_{j=0}^{p} (\omega_3 \|e^{L\Delta t_j} \phi(\mathbf{x}(t_0)) - \phi(\mathbf{x}(t_j))\|^2 + \omega_4 \|\mathbf{x}(t_j) - \psi(\mathbf{z}(t_j))\|^2) \tag{9}$$

For detailed description of each term, please see Appendix A.1-A.2.

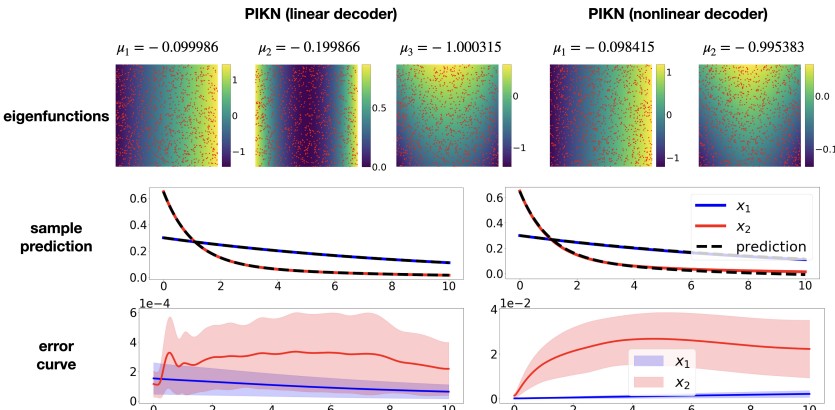

Figure 1: PIKN for autoencoder with linear (left column) and nonlinear (right column) decoder. The first row shows the eigenpairs (red dots are the collocation points); the second row shows a 10000-step forward prediction; the third row visualizes the mean absolute error over 1000 trajectories.

## 3 EXPERIMENTS

### 3.1 SIMPLE NONLINEAR SYSTEM WITH DISCRETE SPECTRUM

First, we consider a simple nonlinear system with a discrete eigenvalue spectrum:

$$\dot{x_1} = \mu x_1, \qquad \dot{x_2} = \lambda(x_2 - x_1^2) \tag{10}$$

For $\lambda < \mu < 0$, this example is serves as a benchmark for Koopman related algorithms because it provides explicitly defined three-dimensional Koopman invariant subspace with $\mu, 2\mu, \lambda]$ and $\phi(\mathbf{x}) = [x_1, x_1^2, x_2 - bx_1^2]$ as Koopman eigenvalues and corresponding eigenfunction, respectively and $b = \frac{\lambda}{\lambda - 2\mu}$. As shown in Fig 1, PIKN is able to find three-dimensional Koopman invariant subspace. Note that these networks are trained in a purely physics-informed manner, indicating there is no need for simulation data. Refer tp Appendix A.3.1 for more details on experiments.

### 3.2 NONLINEAR PENDULUM WITH NO FRICTION

We then study the nonlinear pendulum which exhibits a continuous eigenvalue spectrum with increasing energy:

$$\ddot{x}(t) = -\sin(x) \implies \begin{cases} \dot{x_1} = x_2 \\ \dot{x_2} = -\sin(x_1) \end{cases} \tag{11}$$

Since this example is broadly tested among most of alternative architectures, we use it to benchmark our method. For more setup details, refer to Appendix A.3.2.

All physics-informed counterparts perform better than the original architectures, with PiFBDAE-cont to be the best of all.

### 3.3 HEAT EQUATION

We finally consider the one-dimensional heat equation, as an exmplar for PDEs:

$$u_t = u_{xx}, \quad x \in (-\pi, \pi) \tag{12}$$

with periodic boundary conditions. Using Fourier transform, it can be shown that discrete-time eigenvalues are $\mu_k = -k^2, \quad k = 0, \pm 1, \pm 2, \dots$. To approximately represent the trajectories $u$ and the collocations $u_t$, we discretize the spatial domain with $n = 64$ equally spaced grids. Therefore, we expect our network to at least mimic a discrete Fourier transform and its inverse transform, identifying the right eigenvalues after training. More specifically, we compare the results obtained from the networks that is purely physics-informed, purely data-driven and a mixture of both which we call it hybrid model. We use sums of harmonic functions with random coefficients as collocations

| Model | Train Time | Test MSE |
|---|---|---|
| DAE-disc-latent4 | $299.714s \pm 0.386s$ | $0.019 \pm 0.037$ |
| PiDAE-disc-latent4 | $300.419s \pm 0.524s$ | $0.018 \pm 0.033$ |
| DAE-disc-latent16 | $686.447s \pm 15.906s$ | $0.057 \pm 0.051$ |
| PiDAE-disc-latent16 | $695.323s \pm 2.196s$ | $0.056 \pm 0.049$ |
| DAE-cont | $350.918s \pm 1.101s$ | $0.006 \pm 0.010$ |
| PiDAE-cont | $350.578s \pm 1.702s$ | $0.001 \pm 0.002$ |
| FBDAE-cont | $353.453s \pm 3.005s$ | $0.003 \pm 0.005$ |
| PiFBDAE-cont | $350.975s \pm 1.279s$ | $0.000 \pm 0.001$ |

Table 1: Train time vs prediction on unseen data-sets for different architectures. Methods starts with 'Pi' represents 'physics-informed' are therefore our methods. The number after latent represents the latent dimension of that architecture, for example, DAE-disc-latent2 means a DAE with a $2 \times 2$ matrix $L$. DAE-discTakeishi et al. (2017); Morton et al. (2018), DAE-contLusch et al. (2018) and FBDAE-cont Azencot et al. (2020).

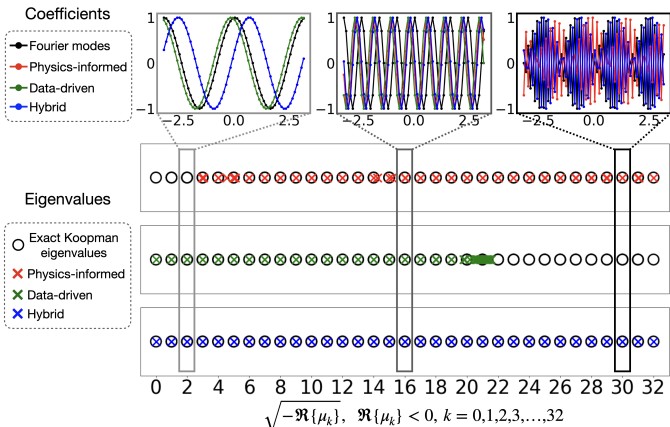

Figure 2: The eigenvalues (with negative real part) of the matrix $L$ from different neural networks are plotted along with the exact, discrete-time eigenvalues of the heat equation at the bottom. The top row shows the coefficients of the linear transformation corresponding to the selected eigenvalues.

for which the analytic spacial derivatives are available. More details of the experimental setup can be found in the Appendix A.3.3. As shown in Fig 2, the transformation coefficients collide with the frequencies of the corresponding Fourier modes. The phase difference is expected because Discrete Fourier transformation is not unique for diagonalization of the heat equation. The networks nearly identify all the correct eigenvalues of the heat equation, however, the purely physics-informed network fails to discover the low-frequency modes. On the contrary, the purely data-driven network misses the high-frequency modes; the hybrid model presents the most satisfying accuracy among all.

## 4 DISCUSSION AND CONCLUSION

In this work, we presented an effective deep learning framework for identifying Koopman eigenvalue and eigenfunction pairs for nonlinear dynamics. In order to validate our method, we carefully went through three examples. In first example, we found the Koopman eigendecomposition only using the knowledge of the system (no data). In second example, we demonstrated adding physics informed loss can improve various autoencoder-based methods. In third example, we show the strength of a hybrid method for a PDE with known solution for the Koopman operator.

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

# A APPENDIX

## A.1 DETAILS OF PIKN TRAINING

In physical sciences, data is scarce while governing equations are available in literature. In physics-informed Koopman networks (PIKNs), we aim to leverage such knowledge of the dynamics, e.g., of Equation 6, to enforce the linearity constraint. The basic idea is to train the network by minimizing the quantity $\|\nabla \varphi_k(\mathbf{x}) \cdot \mathbf{f} - \mu_k \varphi_k(\mathbf{x})\|, \ \forall k = 1, 2, 3, \cdots, M$. More generally, a squared matrix $L$ is used to approximate the Lie operator $\mathcal{L}$, which in turn is related to the Koopman operator, and we minimize $\|L\phi(\mathbf{x}) - \nabla\phi(\mathbf{x}) \cdot \mathbf{f}\|$. Finding the eigenvalue and eigenfunction pairs of the Lie operator corresponds to performing eigendecomposition to the matrix $L$.

### A.1.1 FOR ODE

We first consider an ordinary differential equation. We start from sampling a set of collocation points $X := \{\mathbf{x}_1, \mathbf{x}_2, \cdots, \mathbf{x}_N\}$. This set of collocation points does not need to come from any trajectories of the dynamics but they can be sampled randomly, avoiding a bulk of simulations or measurement data collections.

The objective of the network is to identify a few key coordinates $\mathbf{z} = \phi(\mathbf{x})$ spannned by a set of Koopman eigenfunctions $\varphi_k(\mathbf{x}) : \mathcal{R}^n \to \mathcal{R}, \ k = 1, 2, \cdots, M$ along with a dynamical system $\dot{\mathbf{z}} = L\mathbf{z}$. Objective function is

$$\mathcal{J}_{linear} = \frac{1}{N} \sum_{i=1}^{N} (\omega_1 \|L\phi(\mathbf{x}_i) - \nabla\phi(\mathbf{x}_i) \cdot \mathbf{f}(\mathbf{x}_i)\|^2 + \omega_2 \|\mathbf{x}_i - C\mathbf{z}_i\|^2) \tag{13}$$

if the decoder is linear (where $C$ represents the reconstruction coefficients), or for generic decoder

$$\mathcal{J}_{nonlinear} = \frac{1}{N} \sum_{i=1}^{N} (\omega_1 \|L\phi(\mathbf{x}_i) - \nabla\phi(\mathbf{x}_i) \cdot \mathbf{f}(\mathbf{x}_i)\|^2 + \omega_2 \|\mathbf{x}_i - \psi(\mathbf{z_i})\|^2) \tag{14}$$

Here, $\omega_1$ and $\omega_2$ are the weights for each loss term. The first term encourages linear dynamics within the latent space and the second term makes it a valid auto-encoder. One can further diagonalize $L$ such that the diagonal elements approximate the Koopman eigenvalues and the corresponding outputs of the encoder approximate the Koopman eigenfunctions, respectively. But this constraint is not necessary as it is equivalent to performing eigendecomposition of a general-structured $L$ after training. Once trained, it can be used for state predictions beyond the time horizon used in training.

### A.1.2 FOR PDE

For application to partial differential equations of the form

$$\mathbf{u}_t = \mathbf{f}(\mathbf{u}, \mathbf{u_x}, \dots) \tag{15}$$

similar to ODEs, the goal is to seek coordinates $\mathbf{v} = \phi(\mathbf{u})$ that has linear evolution $\dot{\mathbf{v}} = L\mathbf{v}$ and can be used to reconstruct the original measurements $\hat{\mathbf{u}} = \psi(\mathbf{v})$. The main difference is, however, the input and output of the network are functions of spatio-temporal variables $\mathbf{u}(\mathbf{x}, \mathbf{t})$ instead of the temporal variables $\mathbf{x}(t)$ as in the ODE cases. Therefore, we need to sample a set of "collocation points" in an appropriate function space where $\mathbf{u}_t$ can be cheaply evaluated. We provide examples of such suitable function families in the following chapters.

## A.2 DATA INTEGRATION

Like PINNs, we can seamlessly integrate information from measurement data. Suppose we have snapshots of measurements $X_{data} := \{\mathbf{x}(t_0), \mathbf{x}(t_1), \cdots, \mathbf{x}(t_p)\}$ for an ODE system or $U_{data} :=$

$\{\mathbf{u}(\mathbf{x}, t_0), \mathbf{u}(\mathbf{x}, t_1), \cdots, \mathbf{u}(\mathbf{x}, t_p)\}$ for a PDE, by adding extra loss terms

$$
\begin{aligned}
\mathcal{J}_{data} &= \frac{1}{p} \sum_{j=0}^{p} (\omega_3 \| e^{L \Delta t_j} \phi(\mathbf{x}(t_0)) - \phi(\mathbf{x}(t_j)) \|^2 + \\
&\qquad \omega_4 \| \mathbf{x}(t_j) - \psi(\mathbf{z}(t_j)) \|^2) \quad \text{(for ODE)} \\
\mathcal{J}_{data} &= \frac{1}{p} \sum_{j=0}^{p} (\omega_3 \| e^{L \Delta t_j} \phi(\mathbf{u}(\mathbf{x}, t_0)) - \phi(\mathbf{u}(\mathbf{x}, t_j)) \|^2 + \\
&\qquad \omega_4 \| \mathbf{u}(\mathbf{x}, t_j) - \psi(\mathbf{v}(\mathbf{u}(\mathbf{x}, t_j))) \|^2) \quad \text{(for PDE)}
\end{aligned}
\tag{16}
$$

we can penalize the network predictions that do not match the real measurements, where $\Delta t_j = t_j - t_0, \ \forall j = 1, 2, \cdots, p$. Again, the first term is the linearity loss and the second term is the reconstruction loss. It should be noted that Eq. 16 is consistent with previous literature Takeishi et al. (2017); Morton et al. (2018); Lusch et al. (2018) on using autoencoders to find approximation of Koopman eigenfunctions and can be seen as a baseline on how physics-informed loss improve the performance of PIKN, although we use several other data-driven models in section 3.2.

## A.3 EXPERIMENTAL SETUPS

For all networks in the experiments, $L$ is set to a zero matrix initially and all other parameters are randomly initialized using the default initializer of Pytorch. The exponential linear unit (ELU) is used as the nonlinear activation function, unless stated otherwise, as we expect the transformations in PIKN to be relatively smooth.

### A.3.1 SIMPLE NONLINEAR SYSTEM WITH DISCRETE SPECTRUM

In this example, we have trained two different types of PIKN: a PIKN with a linear decoder and a PIKN with a nonlinear decoder. In both experiments, 1000 collocation points were uniformly sampled from $[-1, 1] \times [-1, 1]$ for training, an Adam optimizer with a learning rate of $1e-4$ has been applied. The total number of training epochs is set to 50000 and the weights in the loss function are set to $\omega_1 = \omega_2 = 1$. The encoder part of both architectures are the same: a 2-layer fully-connected neural network with hidden layer containing 50 neurons. The major differences between the two architectures are as follows:

- The linear decoder is simply a linear layer without a bias term whereas the nonlinear decoder is symmetric to the encoder: a 2-layer fully-connected neural network with hidden width 50.

- For the PIKN with a linear decoder, we have a three-dimensional latent space whereas for the PIKN with a nonlinear decoder, it is two-dimensional.

### A.3.2 NONLINEAR PENDULUM

In this example, we implement three different types of architectures from existing literature and benchmark against their physics-informed counterparts (our methods). The first type of architecture assumes a discrete eigenvalue spectrum of Koopman operator (DAE-disc) and has different versions of implementations Takeishi et al. (2017); Morton et al. (2018). The main difference is on how we obtain the matrix $L$ of the linear dynamics. We follow the latter and treat $L$ as a trainable parameter in Pytorch. We also experiment with different latent dimensions (2, 4, 8, 16) to study the effects. The second type of architecture is proposed by Lusch et al. (2018) which uses an auxiliary network to parameterize the continuous spectrum (DAE-cont), the matrix $L$ varies according to the inputs which effectively captures the frequency shifts. The third type further improves upon it by considering the inverse dynamics (FBDAE-cont). The consistency between forward and backward dynamics is enforced by an extra loss term Azencot et al. (2020). In our implementations, decoder is linear, encoder is a 3-layer fully connected neural network with hidden width 20 and the auxiliary network is a 4-layer fully connected network with hidden width 10.

For training, 1000 collocation points and 20 initial conditions are uniformly sampled from $[-1, 1] \times [-1, 1]$. The initial conditions are further used to generate sequence of snapshots with a temporal

| | PIKN(linear decoder) | PIKN(nonlinear decoder) |
|---|---|---|
| **Experiment 1** | $\mu_1 = -0.10000689$, $\mu_2 = -0.19914131$, $\mu_3 = -0.9996788$ | $\mu_1 = -0.09756267$, $\mu_2 = -0.99973810$ |
| **Experiment 2** | $\mu_1 = -0.10009335$, $\mu_2 = -0.19947967$, $\mu_3 = -1.0004123$ | $\mu_1 = -0.09644943$, $\mu_2 = -0.99867420$ |
| **Experiment 3** | $\mu_1 = -0.10005733$, $\mu_2 = -0.19892442$, $\mu_3 = -1.0003881$ | $\mu_1 = -0.09592330$, $\mu_2 = -0.99915651$ |
| **Experiment 4** | $\mu_1 = -0.10008135$, $\mu_2 = -0.20019206$, $\mu_3 = -0.9986593$ | $\mu_1 = -0.09784412$, $\mu_2 = -1.00298023$ |
| **Experiment 5** | $\mu_1 = -0.09990316$, $\mu_2 = -0.20018657$, $\mu_3 = -0.9991975$ | $\mu_1 = -0.09837312$, $\mu_2 = -1.00187280$ |
| **Experiment 6** | $\mu_1 = -0.09997816$, $\mu_2 = -0.19966313$, $\mu_3 = -1.0005931$ | $\mu_1 = -0.09764695$, $\mu_2 = -1.00517617$ |
| **Experiment 7** | $\mu_1 = -0.10009032$, $\mu_2 = -0.19991782$, $\mu_3 = -1.0000535$ | $\mu_1 = -0.09866422$, $\mu_2 = -0.99622254$ |
| **Experiment 8** | $\mu_1 = -0.09998395$, $\mu_2 = -0.19948480$, $\mu_3 = -0.9996783$ | $\mu_1 = -0.09856838$, $\mu_2 = -1.00204348$ |
| **Experiment 9** | $\mu_1 = -0.09996726$, $\mu_2 = -0.19905518$, $\mu_3 = -1.0005406$ | $\mu_1 = -0.09929386$, $\mu_2 = -0.99679565$ |
| **Experiment 10** | $\mu_1 = -0.10004932$, $\mu_2 = -0.19994377$, $\mu_3 = -0.9993069$ | $\mu_1 = -0.09756234$, $\mu_2 = -1.00279380$ |
| **summary** | $\boldsymbol{\mu_1 = -0.10 \pm 6.22e-04}$, $\boldsymbol{\mu_2 = -0.20 \pm 4.37e-04}$, $\boldsymbol{\mu_3 = -1.00 \pm 5.99e-05}$ | $\boldsymbol{\mu_1 = -0.10 \pm 2.74e-03}$, $\boldsymbol{\mu_2 = -1.00 \pm 9.68e-04}$ |

Table 2: Eigenvalues identified by PIKNs at all training runs. The first column represents the PIKN with a linear decoder and the second column is the one with a nonlinear decoder.

gap $\Delta t = 0.1$ for 10 steps. An Adam optimizer with a learning rate of $1e-3$ has been applied. The total number of training epochs is set to 20000.

For testing, another 10 initial conditions are sampled from $[-1, 1] \times [-1, 1]$ and snapshots are generated with a temporal gap $\Delta t = 0.001$ for 1000 steps. The smaller $\Delta t$ ensures a more accurate forward-stepping with time-varying matrix $L$ in the latter two types of architectures.

### A.3.3 HEAT EQUATION

In the PIKN architecture for Heat equation, encoder and decoder are both linear and the latent dimension is set to 64, the same as the number of spatial grids. Number of training epochs is 100000 and an Adam optimizer has been applied for the training algorithm. In this set of experiments, we use an adaptive learning rate: initially set to 0.01, it keeps decreasing by a factor of 0.5 if no improvements are made over the recent 5000 epochs, until it hits the minimal value of 0.000001.

With the network architecture and training parameters fixed, we train it in three different ways. Namely, we set different values for the weights $\omega_1, \omega_2, \omega_3, \omega_4$ in the loss function, leading to three different training regimes:

- $\omega_1 = 0.0001, \omega_2 = 1, \omega_3 = 0, \omega_4 = 0$: the network is purely physics-informed.

- $\omega_1 = 0, \omega_2 = 0, \omega_3 = 1, \omega_4 = 1$: this corresponds to a purely data-driven learning.

- $\omega_1 = 0.0001, \omega_2 = 1, \omega_3 = 1, \omega_4 = 1$: this represents the scenario where we train a physics-informed network with data integration. For simplicity, we call it hybrid training.

Notice that the value of $\omega_1$ is set on a different order compared to other weights, it is because this loss term involves calculation of numerical derivatives which usually has a greater amplitude. This is a well-known issue for physics-informed neural network and has been thoroughly studied. Adaptive re-weighting schemes were proposed to fix it Wang et al. (2021); Maddu et al. (2021); Zubov et al. (2021). In our case, however, we find that choosing a fixed, small value $\omega_1 = 0.0001$ is sufficient

| (#collocations, #snapshots) | $\mu = -0.1$ | $\lambda = -1$ | eigenvalues=(-1, -0.1) |
|---|---|---|---|
| $(0, 1000)$ | – | – | $(-0.962 \pm 0.118, -0.096 \pm 0.282)$ |
| $(250, 750)$ | $-0.099 \pm 0.002$ | $-0.980 \pm 0.033$ | $(-0.974 \pm 0.043, -0.099 \pm 0.001)$ |
| $(500, 500)$ | $-0.099 \pm 0.000$ | $-0.991 \pm 0.005$ | $(-0.985 \pm 0.010, -0.099 \pm 0.001)$ |
| $(750, 250)$ | $-0.099 \pm 0.003$ | $-0.972 \pm 0.043$ | $(-0.951 \pm 0.073, -0.096 \pm 0.010)$ |

Table 3: Networks are trained with different combinations of data-sets shown in the first column. The first row represents data-driven training and others are hybrid PIKN. The identified system parameters $\mu$ and $\lambda$ and eigenvalues of Koopman operator are shown, respectively, in second, third, and fourth column.

for achieving a fast convergence[1].

For the physics-informed learning, we create 1000 trial functions $\{u^{(1)}, u^{(2)}, \ldots, u^{(1000)}\}$ for training purpose. Each trial function $u^{(j)}$ is a superposition of the Fourier modes that satisfy the periodic boundary conditions. In our case, this amounts to using $\sin(kx)$ and $\cos(kx)$ for $k = 0, 1, 2, \ldots$ as basis functions. The value of $k$ is restricted to be no greater than 32 due to the choice of our grid spatial resolution. The spatial derivatives of the trail functions $\{u_{xx}^{(1)}, u_{xx}^{(2)}, \ldots, u_{xx}^{(1000)}\}$ are calculated using numerical spectral method.

In the data-driven regime, we use simulation data obtained from a solver based on spectral method. We run the simulation with 1000 different initial states. For each run, the initial state of $u$ is obtained through the same procedure as we obtain the trial functions $u^{(k)}$. Then we sample snapshots of the state $u$ with a temporal gap $\Delta t = 0.01$ for 5 steps (i.e. $p = 5$).

For the hybrid training regime, both of the above two data-sets are used. However, to study the effects of the amount of simulation data, we conduct 10, 50, 100, 500, 1000 simulation runs in 5 separate groups of experiments.

### A.3.4 HARDWARE

All experiments were computed on a `slurm`-allocation that had 2 CPUs of an `Intel(R) Xeon(R) CPU E5-2630 v4 @ 2.20GHz`, 16 GB of memory, and one `Tesla K80` GPU. The experiments were implemented using `Python 3.9.12` and `PyTorch 1.11.0` that was using the GPU for compute.

## A.4 ADDITIONAL DISCUSSIONS

### A.4.1 POTENTIAL PITFALLS OF USING NONLINEAR TRANSFORMATIONS

For the example of ODE with discrete spectrum, we have run the training algorithm for 10 times for both architectures and the results are robust in the sense that the identified eigenvalues are all centered around the real ones we derived analytically, which are presented in Table 2. One can see our PIKNs faithfully recover the desired Koopman eigenvalues. The one with the linear decoding provides slightly more robust results, indicating the benefits of adding more known constraints.

We notice that, however, the PIKN with nonlinear decoder doesn't always identify the eigenvalue $\mu_1 = -0.1$. If we significantly change the initialization of the network parameters, sometimes other values emerge. This indicates other transformations exist to linearize the dynamics and reconstruct the state variables. As an example, one can easily check $\varphi_\mu^\beta = x_1^\beta$ for all $\beta \in \mathbb{N}$ are all valid Koopman eigenfunctions associated with eigenvalues $\mu\beta$ and can be used for reconstruction. The lesson here is that by using a nonlinear decoder, we not only increase the flexibility of the Koopman operator theory framework, but also dramatically increase the searching space, which may lead to different learning outcomes.

### A.4.2 UNKNOWN PARAMETERS IN PHYSICAL SYSTEMS

In this experiment, we demonstrate another advantage of PIKN in comparison to data-driven only approaches. We demonstrate that PIKN can leverage partial knowledge of physics, e.g. when some parameters of the system are unknown and even estimate those missing parameters. To illustrate this, we consider the dynamics of the form of Eq. 10 but we let $\mu$ and $\lambda$ to be unknown parameters, which can be treated as trainable parameters with random initialization. We use different combinations of training data-sets and each model is trained for 10 times with different random initializations for the unknown parameters. The difference of data-sets is related to the combination of snapshots (obtained by simulator) versus collocation points (sampled from appropriate function space with no requirement of simulation). The results are shown in Table 3: all hybrid models successfully identified Koopman eigenvalues with higher accuracy than the data-driven model (first row). Note that, the data driven model does not involve a parameter estimation procedure and thus does not provide any knowledge of the physical parameters. On the other hand, the hybrid models are able to solve for the unknown parameters $\mu$ and $\lambda$ correctly by filling gaps in the knowledge of physics with data, effectively operating as a simple model-discovery tool. The experiment shows that incorporating a physics-informed loss is beneficial even when only partial knowledge of physics is accessible to the practitioner.

### A.4.3 ALTERNATIVE APPROACH TO ENFORCE LINEARITY

In fact, minimizing $\|L\phi(\mathbf{x}) - \nabla\phi(\mathbf{x}) \cdot \mathbf{f}(\mathbf{x})\|$ is not the only way to enforce linearity. Take ODE as an example, the decoder reads

$$\hat{\mathbf{x}}(t) = \psi(\mathbf{z}(t)) \tag{17}$$

This implies

$$\frac{d\hat{\mathbf{x}}}{dt} = \nabla\psi(\mathbf{z}) \cdot (L\mathbf{z}) \tag{18}$$

Therefore, minimizing $\|\nabla\psi(\mathbf{z}) \cdot (L\mathbf{z}) - \mathbf{f}(\hat{\mathbf{x}})\|$ is an alternative way to enforce linearity. This alternative approach is also consistent with 'future state prediction' loss in the work of Lusch et al. (2018). The difference between these two ways is whether the linear constraint is applied to the encoder or decoder. In practice, however, we find using either form of the linearity loss or use both of them all work well and does not lead to significantly different results.

---

[1]Another way to get around this is to learn the pseudo-inverse of $L$ instead. Then the loss term reads $\|\phi(\mathbf{x_i}) - L^\dagger \nabla\phi(\mathbf{x_i}) \cdot \mathbf{f}(\mathbf{x_i})\|$. In that case $\omega_1$ and $\omega_2$ can both be set to 1 because the two loss terms are approximately on the same scale.

