# OpenReview forum: "Physics-Informed Koopman Network for time-series prediction of dynamical systems"
_ICLR.cc/2024/Workshop/AI4DiffEqtnsInSci — AI4DiffEqtnsInSci @ ICLR 2024 Poster_

### Official Review · Reviewer_RjD7 · 2024-02-24
**A new loss for data-driven approximation of the Koopman operator**

**Rating:** 8
**Confidence:** 3

**Review:**

This paper aims to improve the approximation of Koopman operators by introducing a new loss function based on mathematical properties of the infinitesimal generator of the Koopman operator. The paper is clear, and offers a simple, generic alteration for practitioners approximating Koopman operators.

The main pitfall of this paper is the lack of comparison to existing loss functions for the same task. At least one should be included in the main paper, in order to showcase the difference between existing literature and this paper.

Overall, the paper is well-written and makes a compelling case for practitioners to adopt this loss.

Some minor nitpicks:
- Each of the terms of this loss are immediately interpretable by someone with a background in the theory of linear dynamical systems, except for the terms $z$ and $z_i$. Therefore, consider pulling the definition out of the appendix so it's possible to understand the loss without going to the appendix. Especially since this is the central contribution of the paper.
- $\omega$ is used both as parameters in $\psi$, and as weight terms in the loss function, ie. $\omega_1, \omega_2, ...$ Consider using a different notation for these.
- In the third sentence after equation (2), the authors state "If, $f$ is sufficiently smooth, ..." without a reference to the smoothness conditions.
- In the second sentence after equation (6), the authors state "It is straightforward to show that Koopman eigenfunctions $\phi(x)$ that satisfies $\mathcal{K}^t \phi(x) = \lambda^t \phi(x)$ for $\lambda^t \neq 0$ are also eigenfunctions of the Lie operator..." The wording here is not entirely clear, is this to say eigenfunctions which satisfy “for all t there exists $\lambda^t$ such that $\mathcal{K}^t\phi(x) = \lambda^t \phi^t...$?“
- In the paragraph preceding equation (8), it reads "Since our ultimate goal is to study nonlinear dynamical systems using linear theory, we do not need to restrict ourselves to Equation 7." This is again not entirely clear. Why should one not limit themselves to studying linear functions of the Koopman eigenfunctions?
- In the first sentence following equation (10), "$\mu, 2\mu, \lambda]$" is missing an opening bracket.
- In the first sentence of section 3.3, "exmplar" is misspelled.

---

### Official Review · Reviewer_Cif2 · 2024-02-27
**Review of "Physics-Informed Koopman Network for time-series prediction of dynamical systems"**

**Rating:** 3
**Confidence:** 3

**Review:**

# Summary

The paper proposes a physics-inspired soft constraint for training Koopman networks.
The constraint in question follows from the definition of the Koopman operator, and is given by $\mathcal{L}g = \nabla g \cdot \mathbf f$, where $\mathcal{L}$ is the Lie operator, $g$ is the observable function, and $\mathbf f$ is the ground truth ODE given by $\frac{d}{dt}\mathbf x(t) = \mathbf{f}(\mathbf{x}(t))$.
In practice, the Lie operator is approximated by a square matrix $L$, which is learned during training and whose eigenfunctions and eigenvalues can then be estimated.
In experiments on a number of prototypical systems, the proposed method is able to recover the correct eigenfunctions and eigenvalues of the Lie operator.

# Strengths
1. (**significance**) The work has the potential to be impactful, due to apparently widespread interest in obtaining linear representations of nonlinear dynamics.
1. (**significance**) The proposed soft constraint is, to the best of my knowledge, a novel contribution.

# Weaknesses
1. (**significance**, **clarity**) It is not entirely clear what the paper is trying to achieve with the proposed method.
Unlike the data-driven methods referenced in the introduction, such as those based on dynamic mode decomposition (DMD) and variational autoencoders (VAEs), the soft constraint proposed here requires complete *a priori* knowledge of the ground truth dynamics $\mathbf f$.
In other words, the proposed method incorporates the ground truth dynamics into the training procedure.
It is then reasonable to ask: if we already know $\mathbf f$, why would we want to train a Koopman network to learn (a linear representation of) $\mathbf f$?
Despite the title of the paper, it seems that the proposed method is not very interesting in a purely time-series prediction paradigm, since we necessarily already know the ground truth dynamics.
We are therefore left to speculate about other settings in which the proposed method might be useful.
For example, and this is not discussed much in the paper, one might suppose that obtaining a linear representation of some known, nonlinear $\mathbf f$ is a worthwhile aim in and of itself.
According to Brunton and Kutz [1], “Expressing nonlinear dynamics in a linear framework is appealing because of the wealth of optimal estimation and control techniques available for linear systems.”
However, if this were the aim of the method, then this needs to be stated clearly and the present work put into the context of related methods for obtaining the Koopman modes from $\mathbf f$, including purely analytic approaches such as Laurent series expansions [1].
Indeed, while all of the systems studied in this paper have analytically tractable eigenfunctions and eigenvalues, there may exist systems where they are intractable [2], offering a potentially useful application of the proposed method.
1. (**quality**) The introduction claims that "We demonstrate doing so reduce the need for large training data-sets and it enables the model to better predict beyond the training horizon," but neither of these claims is backed up clearly by the results.
The first claim would require a comparison to related methods while using varying amounts of training data.
The second claim would require a comparison to related methods using some kind of forecasting metric such as relative error (but I am still not convinced that this would be an interesting application of this method, as discussed in the previous point).
1. (**quality**, **clarity**) Further to the previous point, Section 3.2 claims that "All physics-informed counterparts perform better than the original architectures", yet the results in Table 1 suggest no significant difference between the proposed method and the baselines (although the exact meaning of the error quantity, i.e. the number after $\pm$, is not stated anywhere).
2. (**clarity**, **completeness**) There is no discussion in the paper about how to choose the dimension of the Koopman invariant subspace when it is not known *a priori*.
3. (**clarity**) Unfortunately, there are issues throughout the text with spelling, punctuation, formatting, and citations.
As a result, presentation is a serious issue for the paper in its current form, which is hard to follow in places.
To give some concrete examples:
    - The citations at the beginning of the introduction are not formatted correctly.
    - Symbols and subscripts in the methods section are not well defined or consistent.
    For example, in Section 2.2, $p$ is used to mean both the number of measurements and the number of collocation points (we have to go to the Appendix to find out that $N$ is in fact the number of collocation points).
    In the same section, $\mu_k$ appears out of nowhere, which is presumably supposed to be $\lambda_k$.
    - Table 1 is not referenced anywhere in the text, although it apparently contains the results for Section 3.2.

# Conclusion
While the paper has serious issues in its current form, I think most of them can be addressed fairly easily.
In particular, with regard to the first weakness discussed above, I would encourage the authors to think carefully about what problem is solved by the proposed method and how it relates to existing work.

# Citations
[1] Brunton SL, Kutz JN. Data-Driven Science and Engineering: Machine Learning, Dynamical Systems, and Control. Cambridge University Press; 2019.

[2] Lusch, B., Kutz, J.N. & Brunton, S.L. Deep learning for universal linear embeddings of nonlinear dynamics. Nat Commun 9, 4950 (2018).

---

### Meta-Review · Area_Chair_nw6p · 2024-02-28

**Recommendation:** Accept (Poster)

**Metareview:**

Dear Authors,

Thank you for submitting the draft.

Both reviewers agree that the presented work presents some interesting strengths. However, both reviewers do also raise some major points of concern, especially regarding the clarity of the presentation and some of the claims made. It is expected that authors will be addressing comments by the reviewers in the final draft.

regards

AC

---

### Decision · Program_Chairs · 2024-02-29

Accept (Poster)